# Comparison of heart rate obtained from shorter duration Holter recordings to 24-hour mean heart rate in dogs with atrial fibrillation

**Tamilselvam Gunasekaran**[1], **Bari Olivier**[1], **Lucas Griffith**[2], **Robert Sanders**[1]*

**1** Department of Small Animal Clinical Sciences, College of Veterinary Medicine, East Lansing, Michigan, United States of America, **2** Ames Laboratory, U.S. Department of Energy, Iowa State University, Ames, Iowa, United States of America

* ras@msu.edu

## Abstract

The objective of this study was to evaluate the accuracy of short duration electrocardiographic (ECG) recordings extracted from ambulatory continuous ECG (Holter) to assess 24-hour mean heart rate in dogs with atrial fibrillation. In this retrospective study, Holter recordings obtained from 20 dogs with atrial fibrillation were selected for analysis. Ten out of 20 dogs were receiving drugs to control heart rate at the time of Holter evaluation. From the Holter recordings, heart rate averages were calculated for various sample durations (five-minutes, 30 minutes, one-hour, two-hours, and three-hours) for each dog. Percentage of these shorter duration ECG obtained HR averages that fell within ±10%, ±15% and ± 20% of 24-hour mean heart rate was determined for each sample duration and for each dog. Seventy five percent of heart rate averages obtained from three-hour ECG recordings fell within ±10% of 24-hour mean HR. All the heart rate averages obtained from two-hour ECG recordings fell within ±20% of 24-hour mean heart rate. Based on the results of this study it can be concluded that the duration of the ECG recording affects the prediction accuracy for 24-hour Holter mean HR. Only two and three hours of Holter recordings provided all heart rate averages within ±20% of 24-hour mean heart rate. No significant differences were noted in the prediction accuracy of shorter duration ECG recordings based on rate control therapy status. Further prospective studies are needed to assess the accuracy of HR obtained at home using various ECG recording devices to predict 24-hour mean heart rate in dogs with atrial fibrillation.

## Introduction

Atrial fibrillation (AF) is one of the most common pathologic supraventricular tachyarrhythmias noted in dogs [1]. It frequently occurs with atrial enlargement secondary to an underlying cardiac disease but can also occur without any evidence of structural heart disease (lone AF or idiopathic AF) [2]. Atrial fibrillation is characterized by rapid and irregular atrial activity resulting in loss of atrial contribution to the ventricular filling and reduced cardiac output [3]. Additionally, long-term persistence of rapid and irregular ventricular response rate can

**Competing interests:** The authors have declared
that no competing interests exist.

deteriorate cardiac function and worsen hemodynamics [3]. Management of AF involves
either cardioversion of AF rhythm into sinus rhythm or ventricular rate control using pharma-
cological agents [4–8]. Electrical cardioversion has been shown to be safe and effective in dogs,
but recurrence of AF is common, particularly in dogs with underlying structural heart disease
[5, 6]. Medical cardioversion using amiodarone has been reported in dogs but systemic side
effects with long-term amiodarone use remain a concern [7]. Consequently, pharmacological
HR control through reduction in atrio-ventricular nodal conduction is the most commonly
used treatment modality for management of AF in dogs [4].

Various methods can be used to evaluate heart rate (HR) in dogs with AF including cardiac
auscultation, pulse assessment, short-term in hospital ECG and 24-hour continuous ambula-
tory electrocardiography (Holter). A previous study in dogs with AF has shown that HR
obtained by cardiac auscultation is inaccurate and cannot be relied upon for management of
AF [9]. Additionally, pulse assessment for HR estimation is likely to be inaccurate due to fre-
quent pulse deficits noted in dogs with AF. Short-term in-hospital ECG recordings are com-
monly used to evaluate the need for rate control therapy and to adjust medications for HR
control in dogs with AF. However, HR obtained by short-term, in-hospital ECG may not rep-
resent the achieved HR control at home. It has been shown that HR obtained by the one-min-
ute in-hospital ECG overestimated the 24-hour mean HR by 26 bpm compared to a 24-hour
Holter recording in dogs with AF [10]. Hospital environments might cause an increase in
autonomic tone and consequently result in a higher HR in dogs [10, 11]. This could result in
inappropriate changes in therapy, potentially impacting control of clinical signs and
morbidity.

A current recommendation is to perform 24-hour Holter monitoring to assess HR in dogs
with AF [4]. Twenty-four-hour Holter recording provides a more complete representation of
HR pattern in the home environment and provides information about the presence of any
concurrent arrhythmias that may influence treatment decisions. Additionally, 24-Holter
recording provides prognostic information [12]. A recent retrospective study in dogs with AF
has shown that dogs with a mean 24-hour HR above 125 bpm had a shorter survival time com-
pared to dogs with lower 24-hour mean HR averages [12]. Despite the benefits of 24-hour Hol-
ter recordings, the use of Holter monitoring may be limited by potential disadvantages such as
cost, recording duration and owner compliance. Therefore, there is a need to identify a practi-
cal and cost-effective alternative to 24-hour Holter monitoring to estimate mean HR in dogs
with AF. Short-term ECG recordings performed at home, unaffected by the stress of hospital
visits, might provide an effective alternative to long term ambulatory ECG recording. Such
short duration ECG recordings at home must be long enough to reflect the dog's 24-hour
mean HR and be unaffected by sporadic variations in HR. There are several at home HR moni-
toring tools (i.e. smartphone based single lead ECG devices, cardiac loop recorders (implant-
able and cutaneous), wearable HR monitoring technology that uses photoplethysmography or
other pulse sensing technology) commercially available that can be used for HR assessment in
dogs with AF. In general, the accuracy of non-ECG based technologies to estimate HR in AF
has not been evaluated in dogs. Moreover, methods of HR estimation based on pulse detection
or peripheral rate of blood flow are unlikely to be accurate given the frequent nature of pulse
deficits in dogs with AF and requires further investigation. ECG based devices such as a smart-
phone-based ECG (AliveCor®, AliveCor® Inc., San Francisco, California) has been previ-
ously evaluated in dogs with AF and was noted to have good owner compliance in using these
devices for short duration ECG recording [13]. While AliveCor smart phone ECG provided
good rhythm evaluation in dogs, there were considerable differences between veterinary cardi-
ologists for estimation of HR using AliveCor obtained ECG in dogs with both sinus rhythm
and AF [13]. More importantly, the accuracy and adequacy of short-term ECG recordings that

can be obtained using any device at home to estimate 24-hour mean HR in dogs with AF is not known. The goal of this retrospective study was to assess the HR averages obtained using shorter duration ECG recordings (five-minutes, 30 minutes, one hour, two hours and three hours of ECG recordings extracted from a Holter) for their deviation from 24-hour mean HR in dogs with AF.

## Materials and methods

### Animals

Medical records of dogs evaluated at the Veterinary Medical Center, College of Veterinary Medicine, Michigan State University between the years 2005–2012 and diagnosed with AF were retrospectively reviewed. Only dogs that subsequently underwent Holter monitoring as part of their clinical evaluation were included in further analysis. Holter recordings were only selected for further analysis if no changes were made to the current rate control therapy or no new antiarrhythmic drugs were initiated at the beginning Holter monitoring. Holter monitoring was performed using two or three channel recorders (Lifecard CF, Spacelab's Healthcare, Snoqualmie, Wisconsin, United States) as previously described for Holter application in dogs [14]. The dogs were sent home for Holter monitoring and the owners were asked to maintain a record of normal daily activities and note any clinical signs.

### Holter data analysis

The Holter recordings were analyzed using commercially available analysis software (Pathfinder SL, Spacelab's Healthcare, Snoqualmie, Wisconsin, United States). The entire recording was visually inspected by a cardiology resident (TG) under the supervision of a veterinary cardiologist (RAS) and corrected for any automated errors. After manual verification, continuous RR interval data (defined as the distance between two consecutive R waves measured in milliseconds) for the entire Holter recording was extracted for each dog. This data was then edited such that analysis of R to R intervals began at midnight and ended twenty-four hours later to provide a uniform 24 hours of continuous RR interval data for all dogs. For example, if Holter monitor was placed on a dog at 3 pm during the day, from 3 pm to midnight the RR interval data was discarded. Discarding the initial portion of the Holter recording in this way provided a uniform, time adjusted, at home, continuous RR interval data for all dogs. Dogs were excluded if the edited recording duration was less than 24 hours, if there was more than five percent artifact in the edited recording or if more than two percent ventricular aberrancy was noted. From the 24 hours of continuous RR interval data, 24-hour mean HR was calculated for each dog. Heart rate averages were calculated for sample durations of five minutes, 30 minutes, one hour, two hours, and three hours from the continuous RR interval data for each dog using a commercial software (MATLAB and Simulink software, MathWorks, Torrance, California, United States). This resulted in 288, five-minute HR averages, 48, thirty-minute HR averages, 24, one-hour HR averages, 12, two-hour HR averages and eight, three-hour HR averages available for analysis for each dog.

### Statistical analysis

Statistical analysis was performed using a commercially available statistics software (Stata, StataCorp LP, College Station, Texas, United States). Statistical significance for a type-1 error was set at <0.05. Normality was tested using Kolmogorov-Smirnov test. Categorical data was represented as number of observations and percentages. Normally distributed continuous variables were presented by mean and standard deviation. Non-normally distributed continuous

variables were presented by median and range. The unpaired Student's t test was used to compare differences in mean HR based on rate control therapy status. From the calculated HR averages for each shorter sample duration, percentage of HR averages that are within ±10%, ±15% and ±20% of 24-hour mean HR for each dog was calculated. The Wilcoxon rank sum test was used to assess the differences in the percentage of short duration HR averages that fell within a specified range from the 24-hour mean HR based on rate control therapy status. The bootstrap method was used to construct 95% confidence intervals (percentile method) for the median percentage of HR averages failing within specific percentage of the 24-hour mean HR (5000 repetitions with replacement) as previously described [15]. Linear mixed model for repeated measures using 24-hour mean HR as an independent variable, percentage of HR averages that are within ±10% of 24-hour mean HR as the dependent variable and individual dog as random effect was modelled (random intercept model). To achieve an interpretable intercept for the model, 125 was deduced from each individual dogs 24-hour mean HR.

## Results

### Demographics

Entire study population consisted of 20 dogs with AF who underwent Holter monitoring. The median age was eight years (range = 5.0–12.0 years) for all the dogs. Twelve dogs were males (10 / 12 neutered) and eight dogs were females (8 / 10 spayed). The median body weight was 38.2 kg (range = 12.9–78.6 kg). The breed representation is as follows, Great Dane (3), Bulldog (1), Bullmastiff (2), Rottweiler (1), Newfoundland (1), Airedale (1), Scottish Deerhound (1), Keeshond (1), Golden Retriever (3), Labrador retriever (1), Irish Wolfhound (1), and mixed breed (4). Nine dogs were diagnosed with dilated cardiomyopathy, eight dogs with myxomatous mitral valve disease, one dog with severe subaortic stenosis, one dog with a left to right shunting patent ductus arteriosus, and one dog was suspected to have lone AF. Ten dogs were not receiving any rate control therapy at the time of Holter evaluation. Ten dogs were receiving rate control therapy for at least 2 weeks at the time of Holter evaluation. Of the ten dogs receiving HR control therapy, six dogs were receiving combination therapy with diltiazem (range 1.1–1.8 mg / kg PO TID) and digoxin (range = 0.003–0.0046 mg / kg PO BID), one dog was receiving combination therapy with diltiazem (1.4 mg / kg PO TID) and atenolol (1.2 mg / kg PO SID), one dog was receiving combination therapy with diltiazem (0.98 mg / kg PO TID) and sotalol (1.3 mg / kg PO BID) and two dogs were receiving monotherapy with diltiazem (range = 1.2–1.78 mg / kg PO TID). Concurrent cardiac medications administered included furosemide (9 / 20), pimobendan (9 / 20), enalapril (4 / 20) and spironolactone (4 / 20).

### Holter results

The unedited Holter recording durations ranged from 32 hours to seven days. Twelve out of 20 (60%) dogs had ventricular premature complexes (range = 25 to 2378 ventricular premature complexes per 24 hour). None of the dogs had > 2% ventricular aberrancy on their 24-hour Holter recording. The 24-hour mean HR was 126 bpm ± 29 for all the dogs in this study. The 24-hour mean HR was not significantly different (p = 0.627) between the dogs that did not receive HR control therapy (134 ± 29 bpm) and dogs that did receive rate control therapy (mean HR = 121 ± 33 bpm). Table 1 depicts the percentage of HR averages falling within ±10%, ±15%, and ±20% of the 24-hour mean HR for a given sample duration in study dogs. Fig 1 depicts the percentage of dogs with all their HR averages falling within ±10%, ±15% and ±20% of 24-hour mean HR. Note that with three hours of continuous ECG recording, 70% of dogs had all their HR averages within ±20% of 24-hour mean HR. No significant differences in

**Table 1. Accuracy of shorter duration ECG recordings in 20 dogs with atrial fibrillation.**

| Sample duration | % of averages within ±10% of 24-hour mean HR (Median (95% CIª) | % of averages within ±15% of 24-hour mean HR (Median (95% CIª) | % of averages within ±20% of 24-hour mean HR (Median (95%CIª) |
|---|---|---|---|
| 5 minutes | 51 (38–62) | 74 (66–79) | 86 (79–88) |
| 30 minutes | 61(39–71) | 81 (70–88) | 93 (81–95) |
| 1 hour | 67 (54–79) | 81 (70–92) | 92 (83–100) |
| 2 hours | 75 (58–83) | 91(83–100) | 100 (96–100) |
| 3 hours | 75 (50–87) | 100 (88–100) | 100 (94–100) |

Table 1 depicts the percentage of mean heart rate averages that fall within ±10%,±15%, and ±20% of 24- hour mean heart rate for a given sample duration in dogs with atrial fibrillation. True mean heart rate is the mean heart rate obtained from a 24-hour Holter monitor.

ªCI is boot strapped 95% confidence interval for the median.

[b] HR = heart rate

the accuracy of short duration ECG samples were noted based on rate control therapy status (p = 0.062).

The 24-hour mean HR was a poor predictor (intercept = 60.6, standard error = 5.5, P = < 0.05; slope = 0.1127, standard error = 0.17, P = 0.5246) of percentage of shorter duration heart rate averages that fall within a specified range of 24-hour mean HR after adjusting for random effects of individual dogs. Seventy six percent of the unaccounted residual variances were attributed to random subject variances (random intercept variance = 372, standard deviation = 19.29). This suggests that at a 24-hour mean HR of 125bpm, on an average 60% of the

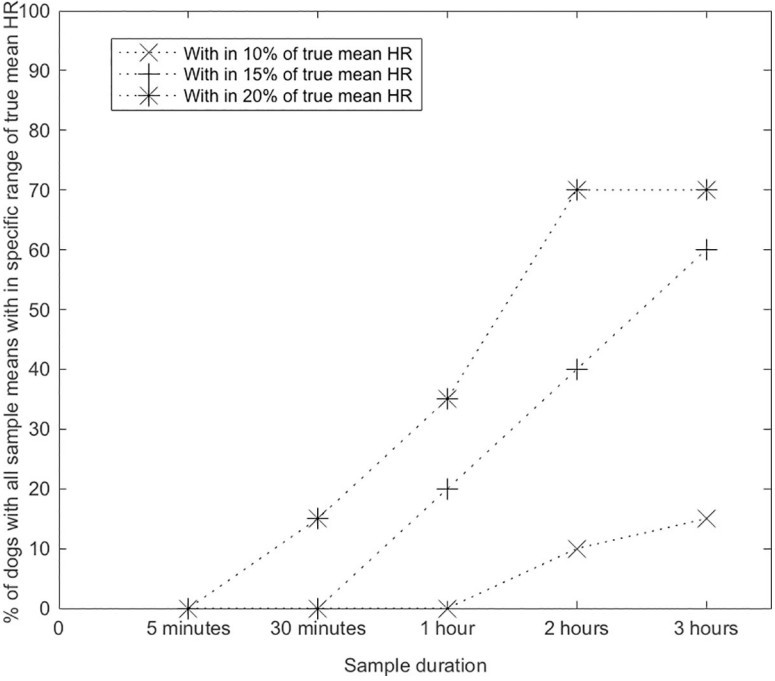

**Fig 1. Line plot of percentage of dogs with all their heart rate averages within a specified range from the 24-hour mean heart rate in 20 dogs with atrial fibrillation.** Fig 1 represents the percentage of dog with all their heart rate averages obtained using shorter duration ECG recordings that fall within ±10, ±15 and ±20% of the true mean heart rate. True mean HR = 24-hour mean HR.

HR averages obtained from short duration ECG recordings fall within ±10% of the 24-hour mean HR, with wide variation in accuracy due to random dog effect.

## Discussion

The results of this study demonstrate that increasing recording durations improve the accuracy of shorter duration ECG recordings to predict 24-hour mean HR in dogs with AF. Also, when a wider accuracy (20%) from 24-hour mean HR is acceptable for a clinical situation, shorter duration ECG recordings may be adequate in providing approximate estimates of 24 hour mean HR. These findings should be interpreted in the context of an individual clinician's judgement on what is an acceptable degree of accuracy for HR evaluation in a dog with AF. For example, if a 20% accuracy from the 24-hour mean HR is acceptable, then a five-minute ECG recording may be reasonable as 86% of the five-minute sample HRs (see Table 1) remain within ±20% of the 24-hour mean HR. If tighter accuracy is desired, such as 10%, then a longer duration ECG recording is necessary. In authors opinion, 20% accuracy from the 24-hour mean HR is likely not acceptable for a dog with a higher 24-hour mean HR, while it may provide a clinically acceptable estimation in a dog with a lower 24-hour mean HR below 100 bpm. However, at the time of initial evaluation of a dog with AF, 24-hour mean HR information is not available to the clinician and it is not possible to know a dog's 24-hour mean HR without Holter evaluations. One exception is in situations of fast atrial fibrillation (as defined by one-minute in hospital obtained ECG HR of more than 155 bpm) the 24-hour mean HR can be predicted to be above 140 bpm which could potentially guide the clinician to make decisions on accuracy and duration of the ECG recording [10]. In addition, due to the presence of large random variation between dogs, it is difficult to predict an acceptable accuracy of a short duration ECG on an individual dog even with if prior 24-hour mean HR is available. Due to the above-mentioned reasons, selection of a specific shorter duration ECG recording for HR evaluation or an acceptable range of accuracy from 24-hour mean HR is difficult at the time of initial evaluation.

### Limitations

The major limitation of this study is the small sample size and retrospective nature of the study. Additionally, there are several other limitations. Most of the study dogs were large breed dogs with only one dog weighing <20kg and only one dog diagnosed with lone AF in this study. These results may not be directly extrapolated to small breed dog population or to dogs with lone AF, but the results do represent the typical clinical demographic of canine AF population. We only used ±10%,±15% and ±20% within 24-hour mean HR as acceptable accuracy from 24-hour mean HR. It is possible that some clinicians accept wider variation from 24 hour mean HR for case management in dogs with AF. Moreover, we only used a maximum of hours of ECG recording to assess the accuracy of shorter duration ECG recordings. To the authors knowledge, currently available at-home tools for continuous, short duration ECG evaluation (including smartphone-based ECG platforms, event recorders or continuous loop recorders) do not have the capacity to record continuous ECG for more than one-hour duration [13, 16]. Therefore, at the time of publication, practically any continuous ECG recording duration longer than one hour may not be practically feasible without placement of a version of Holter monitoring. While some devices have the capacity to record repeated shorter duration ECG recordings (e.g. AliveCor smartphone ECG) the use of such repeated short duration ECG recordings to estimate 24-hour mean HR was not evaluated in this study. Also, this study only evaluated accuracy of shorter duration ECG recordings extracted from a 24-hour Holter. It must be pointed out that results of this study cannot be directly interpreted as the accuracy

of shorter duration ECG recordings obtained using various ECG recording devices that require owner-dog interaction such as placement of ECG recording device on a dog. Such interactions can induce anxiety and may not represent an ECG recording that would be extracted from a Holter. Additionally, the findings of this study do not apply to any non-ECG based HR monitoring systems that have the capacity to record continuous HR information.

Interestingly, there was no significant differences in 24-hour mean HR between dogs that did and did not receive HR control therapy in this study. Particularly, the 24-hour mean HR for the untreated group can be considered low for dogs with AF secondary to structural heart disease. There are several reasons for the lower 24-hour mean HR noted in the untreated group of AF dogs in this study. First, none of the previous studies that reported 24-hour mean HR in dogs with AF used the initial editing process used in this study [10, 12, 17]. Removal of a larger initial portion of the Holter recording may have eliminated high heart rate periods prior to acclimatization of a dog after Holter equipment placement, resulting in lower overall mean HR. Secondly, to the authors knowledge, information about 24-hour mean HR for the untreated group has only been reported previously in two veterinary studies. In one study [10] only a range of 24-hour mean HR was provided for the untreated group, while in the second study [17] dogs with 24-hour mean HR above140 bpm were excluded from the analysis. Therefore, direct comparison to previous studies could not be performed. Finally, in this study, initial Holter evaluation of some untreated dogs provided a mean 24-hour HR that was not high enough to warrant rate control therapy. This resulted in dogs with low 24-hour mean HR to be included in untreated group resulting in lower overall mean HR for the untreated AF group.

## Conclusions

Increasing the duration of a short-term ECG recording improves the prediction accuracy for 24-hour mean HR. The accuracy of each shorter duration samples should be thoroughly considered by the clinician as it has direct impact on the decision to initiate or modify rate control therapy in AF dogs. Inaccuracies of 20% from 24-hour mean HR are likely to over or underestimate the efficacy of rate control therapy for most dogs with AF, thereby impacting treatment decisions and potentially patient outcome. Further prospective studies evaluating the accuracy of short-term ECG recordings during specific times of the day and at times of rest are needed.

## Author Contributions

**Conceptualization:** Tamilselvam Gunasekaran, Bari Olivier, Robert Sanders.

**Data curation:** Tamilselvam Gunasekaran, Lucas Griffith.

**Formal analysis:** Tamilselvam Gunasekaran, Lucas Griffith.

**Investigation:** Tamilselvam Gunasekaran, Bari Olivier.

**Methodology:** Tamilselvam Gunasekaran.

**Project administration:** Tamilselvam Gunasekaran.

**Writing – original draft:** Tamilselvam Gunasekaran, Robert Sanders.

**Writing – review & editing:** Tamilselvam Gunasekaran, Robert Sanders.

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
