## [Decision Letter · Decision Letter 0]

7 Aug 2020

PONE-D-20-23023

Comparison of heart rate obtained from shorter duration Holter recordings to 24-hour mean heart rate in dogs with atrial fibrillation

PLOS ONE

Dear Dr. Sanders,

Thank you for submitting your manuscript to PLOS ONE. After careful consideration, we feel that it has merit but does not fully meet PLOS ONE’s publication criteria as it currently stands. Therefore, we invite you to submit a revised version of the manuscript that addresses the points raised during the review process.

The Authors should respond to all of the Reviewers' comments. One of the major comments of the Reviewers was the potential added benefit of shorter time periods for Holter monitoring over 24 hour monitoring that usually is carried out, due to the fact that animals would still have to undergo the procedures for placing the Holter. Therefore this, especially, would need to be addressed in a revised manuscript. Furthermore, the Reviewers also noted that the statistical analysis of the manuscript should be strengthened.

We look forward to receiving your revised manuscript.

Kind regards,

Daniel M. Johnson, PhD

Academic Editor

PLOS ONE

Journal Requirements:

Reviewers' comments:

Reviewer's Responses to Questions

**Comments to the Author**

1. Is the manuscript technically sound, and do the data support the conclusions?

Reviewer #1: Yes

Reviewer #2: Yes

Reviewer #3: Yes

2. Has the statistical analysis been performed appropriately and rigorously? 

Reviewer #1: No

Reviewer #2: Yes

Reviewer #3: I Don't Know

3. Have the authors made all data underlying the findings in their manuscript fully available?

Reviewer #1: Yes

Reviewer #2: Yes

Reviewer #3: Yes

4. Is the manuscript presented in an intelligible fashion and written in standard English?

Reviewer #1: Yes

Reviewer #2: Yes

Reviewer #3: Yes

5. Review Comments to the Author

Reviewer #1: The authors provide a study that examined whether periods substantially shorter than 24hours of Holter monitoring (ambulatory ECG monitoring) could reasonably accurately reflect the 24 hour average heart rate in dogs with atrial fibrillation.

I have several concerns about the analyses.

First, in previous studies of mean heart rate, investigators suggested trimming the Holter recordings to exclude the first 3 hours and final hour of the recording (https://pubmed.ncbi.nlm.nih.gov/19645836/, see figure 2B) as this removes the effect of stress and anxiety experienced by most dogs when applying the Holter and when returning to have the Holter removed. The residual 20 hours more accurately reflect the true “at home” average heart rate. The authors should consider this approach to see how this affects their interpretation of agreement. They would obviously remove the short periods from these peripheral time periods (because these, too, would be affected by anxiety etc).

The statistical analysis section needs expanding. This reviewer cannot determine what analyses were applied for what purposes. For example, what “study variables” did the authors examine by Spearman Rank Correlation?

The accuracy of short periods might be very dependent on specific dogs. Did the authors account for dog in their analysis? In other words agreement might be high for some dogs and low for others. Examining the % agreement by dog might reveal such a “between dog” effect.

A further approach the authors should consider is to randomly select a 5-minute, 30-minute, 1 hour, 2-hour and 3-hour segment from each dog and see how often these randomly selected segments agree within the predefined limits. This would represent the clinical reality of a client obtaining a heart rate at home.

Minor comments:

Line 88-90. The authors state the limitations of Holters (cost, duration, compliance). However, the study examines Holter recordings and does not provide or discuss what alternatives to Holter monitoring exist or that could substitute Holter monitoring (and this reviewer knows of no such alternatives). Therefore, this statement seems rather contrived and without a solution.

Lines 177-180. If the average rates are not different statistically, then one group cannot have a “slightly higher” rate than the other. Also, a P value should be provided for all statistical testing, whether significant or not. Please amend.

Line 177-180. The average HR for dogs with AF that were not being administered any rate control therapy seem considerably lower than previously documented in dogs with secondary AF. Do the authors have any explanation for this rather low rate? (most previous studies suggest rates exceeding 160-190 bpm). The rates for the group under rate control therapy appears to be consistent with previous observations.

Line 185-186. Could the authors provide scatter plots for each of the short duration recordings vs the Holter recordings plotted as average heart rates? Although the P value for their observation is “not significant”, with only 10 dogs in each group, the probability of this being a false negative is quite large.

Line 204-213. The authors suggest that a 5 minute ECG recording (in-clinic) would mostly agree to within 20% of the Holter-determined average HR. However, this is NOT what the authors showed. To demonstrate that specific idea, the authors would need to examine the 5 minute intervals within the 1st hour of Holter application (while the dog was still in the clinic and stressed) with the average at-home HR. How many of the 240 5-minute intervals obtained during the first hour came within 20% of the average at-home heart rate? If the authors suggest that clients could obtain a 5-minute ECG at home on their pets, the authors would have to suggest how such a feat would be possible, and also demonstrate that doing this would not alter the level of anxiety in the dog during at-home acquisition.

Line 215-220. Several of the “health collars” (e.g. PetPace) can record ECGs in dogs, at least in sinus rhythm. Whether they can accurately count heart rates in dogs with atrial fibrillation remains to be determined. If they can, then they provide continuous monitoring, or can record variable periods for as long as desired (limited by battery life). Therefore, such devices (similar to the iWatch etc, used in humans) are available, but in limited supply.

Reviewer #2: General comments – this manuscript describes a retrospective analysis of subsets of ECG data recorded during a 24 hour Holter recording. While providing interesting preliminary data, I wonder how much less practical difficulty there might be in practice in obtaining shorter Holter recordings compared to a 24 hour recording? The dog still has to have the monitor fitted and wear it for several hours (preferably from the point the dog is relaxed at home, not from the initial time of Holter fitting) – if the benefits are “cost, recording duration and owner compliance” how much difference would shorter recordings truly make? I don’t have a good feel for how much less companies would charge to analyse shorter recordings, unfortunately. The main advantage I can see would be in those cases that the leads became detached after less than 24 hours – if it could be shown that the data was valid if at least 2-3 hours of recording was available once the dog was at home and relaxed that could prevent the need to start all over again. My specific comments, questions and observations are detailed below.

Introduction

Line 62 – should read “…remain a concern.”

Lines 63-65 – this sentence is identical to lines 59-61. I recommend deleting the sentence in lines 59-61, as it makes most sense to keep this information at the end of the paragraph.

Lines 67-79 – you mention that pulse assessment can be used to assess heart rate, but do not discuss later in the paragraph whether this would provide an accurate reflection of heart rate (which seems unlikely, due to pulse deficits, etc.).

Lines 91-93 – how do you propose that the Holter is fitted without “the stress of hospital visits”?

Results

Line 151 – just an idle observation, but I’m surprised only 20 dogs met the inclusion criteria over a 7 year period. Were there a lot of recordings that were <24 hours when edited?

Lines 152-153 and 154-155 – personally, I would just report the range rather than the range and the IQR, as I don’t think the latter adds anything of real value.

Lines 177-180 – please don’t describe non-significant differences. This should simply read “There was no significant difference in heart rate between dogs receiving HR control therapy (121+/- 33bpm) and those not receiving HR control therapy (134 +/- 29 bpm).”

Lines 180-182 – Table 1 and Figure 1 essentially show the same information twice. Please remove either the Table or the Figures.

Lines 182-184 – I’m not sure why this correlation is clinically interesting or important – please explain why you performed this test.

Lines 184-186 – given that rate control status had no significant impact on heart rate, why would you imagine that it would have an impact on accuracy? Again, this seems to be a statistical test that isn’t testing a hypothesis.

Table 1

The information displayed represents “Median (95% CI)” NOT “(Median +/- CI)” – please edit this accordingly.

Discussion

Lines 206-207 – I don’t think that “agreement” changes – you are just being more lenient about what you are willing to accept as “accurate”.

Lines 209-215 – based on your stated mean heart rate of 126 bpm, a 20% inaccuracy gives a heart rate of 151 bpm, which is likely to be interpreted very differently than a mean of 126 by most clinicians. This is clearly potentially problematic – please expand the discussion to reflect this.

Lines 215-220 – this is the first time you have mentioned alternative methods of obtaining an ECG recording in the home environment – which potentially changes ho you might be positioning shorter recordings (although I don’t relish the idea of asking a client to hold an AliveCor against their dog for an hour!). Please add this information to the Introduction (with a brief summary of the work that has already been done looking at accuracy of such recordings in the home environment for monitoring of AF).

Lines 226-228 – is it not equally likely that these would consistently underestimate 24 hour averages (simply because they wouldn’t include the periods of exercise and excitement that also influence mean heart rate over 24 hours)? So you might be at risk of thinking that your rate control was better than it actually is?

Line 236 – please remove the grocer’s apostrophe (the “s” in ECGs denotes that this is plural rather than denoting possession).

Lines 237-248 – I am still not sure why you did this test in the first place (it feels like a fishing trip) so my preference would be to remove this from the discussion.

Lines 248-251 – this is possible, of course, but I think it’s very unlikely (see previous comment regarding lines 209-215).

Lines 255-257 – this would still rely on owners to record these times accurately – is there a way to mitigate this as a potential limitation to future studies?

Conclusions

Lines 260-262 – surely there is experimental evidence that would help determine guidelines for what is acceptable accuracy, rather than simply relying on the judgement of the clinician?

Reviewer #3: The purpose of this study was to determine if the heart rate measured for shorter time periods (5 minutes, one, two and three hours) would equate to the mean rate as determined by 24 hour electrocardiographic monitoring. The mean heart rate for each of these four time intervals was assessed for its ‘closeness’ (within 10%, 15% or 20%) to the 24 hour mean rate. The authors embarked on this investigation with the justification that performing a 24 hour Holter for shorter periods of time would be less problematic for owners and dogs. The conclusion of this paper is that none of the shorter time periods would equate to the mean 24-hour heart rate, but the longer the recording the closer the short surveillance heart rate was to the full recording.

This is a very focused study to determine if short recordings are adequate in the evaluation of rate during atrial fibrillation. The authors state that there are no other methods currently available to determine heart rates in a dog for longer than one hour. However, an Internet search will reveal that individuals are adapting noninvasive monitoring such as Fit Bits for dogs. Additionally, other devices have been developed. (See reference: Yasin Cotur, Michael Kasimatis, Matti Kaisti, Selin Olenik, Charis Georgiou, Firat Güder. Stretchable Composite Acoustic Transducer for Wearable Monitoring of Vital Signs. Adv. Funct. Mater. 2020, 30, 1910288. Doi.org/10.1002/adfm.201910288) These may serve as a way to get minimal information (and likely inadequate when monitoring arrhythmias) regarding heart rate. Such devices could provide data for specific questions such as heart rate. Comparison of these devices with established methods such as 24 hour Holter monitoring would be important to document. It is not clear how this study could be used in such a situation. Authors need to further justify this study.

From an implementation component there are no striking errors in this manuscript. It should be noted however that the average heart rates of the dogs examined are actually low for atrial fibrillation. Dogs that have high ventricular response rates are of course the more critical for monitoring and this paper does not address that kind of patient. Differences between low and high ventricular response rate heart rates over 24 hours is likely meaningful.

6. PLOS authors have the option to publish the peer review history of their article (what does this mean?). If published, this will include your full peer review and any attached files.

Reviewer #1: **Yes: **Mark Rishniw

Reviewer #2: No

Reviewer #3: No

---

## [Author Response · Author response to Decision Letter 0]

26 Aug 2020

PONE-D-20-23023

Comparison of heart rate obtained from shorter duration Holter recordings to 24-hour mean heart rate in dogs with atrial fibrillation

PLOS ONE

Dear Dr. Sanders,

Thank you for submitting your manuscript to PLOS ONE. After careful consideration, we feel that it has merit but does not fully meet PLOS ONE’s publication criteria as it currently stands. Therefore, we invite you to submit a revised version of the manuscript that addresses the points raised during the review process.

The Authors should respond to all of the Reviewers' comments. One of the major comments of the Reviewers was the potential added benefit of shorter time periods for Holter monitoring over 24 hour monitoring that usually is carried out, due to the fact that animals would still have to undergo the procedures for placing the Holter. Therefore this, especially, would need to be addressed in a revised manuscript. Furthermore, the Reviewers also noted that the statistical analysis of the manuscript should be strengthened.

We look forward to receiving your revised manuscript.

Kind regards,

Daniel M. Johnson, PhD

Academic Editor

PLOS ONE

Journal Requirements:

Please find the DOI for the data.

Gunasekaran, Tamilselvam (2020), “Atrial fibrillation dogs”, Mendeley Data, v1http://dx.doi.org/10.17632/hbwy6bn48n.1

Reviewers' comments:

Reviewer's Responses to Questions

Comments to the Author

1. Is the manuscript technically sound, and do the data support the conclusions?

Reviewer #1: Yes

Reviewer #2: Yes

Reviewer #3: Yes

2. Has the statistical analysis been performed appropriately and rigorously? 

Reviewer #1: No

Reviewer #2: Yes

Reviewer #3: I Don't Know

3. Have the authors made all data underlying the findings in their manuscript fully available?

Reviewer #1: Yes

Reviewer #2: Yes

Reviewer #3: Yes

4. Is the manuscript presented in an intelligible fashion and written in standard English?

Reviewer #1: Yes

Reviewer #2: Yes

Reviewer #3: Yes

5. Review Comments to the Author

Reviewer #1: The authors provide a study that examined whether periods substantially shorter than 24hours of Holter monitoring (ambulatory ECG monitoring) could reasonably accurately reflect the 24 hour average heart rate in dogs with atrial fibrillation.

I have several concerns about the analyses.

First, in previous studies of mean heart rate, investigators suggested trimming the Holter recordings to exclude the first 3 hours and final hour of the recording (https://pubmed.ncbi.nlm.nih.gov/19645836/, see figure 2B) as this removes the effect of stress and anxiety experienced by most dogs when applying the Holter and when returning to have the Holter removed. The residual 20 hours more accurately reflect the true “at home” average heart rate. The authors should consider this approach to see how this affects their interpretation of agreement. They would obviously remove the short periods from these peripheral time periods (because these, too, would be affected by anxiety etc).

We did undertake such a measure to remove the initial portion of the Holter recording that is likely affected by the stress of a hospital visit. We removed the initial portion (> 3 hrs in every case) of the Holter recording from the time of Holter application to midnight on the day of Holter application. Therefore, for each dog 24 hours of Holter data starting at midnight to midnight next day was extracted for further analysis. We noted this in 136-137 lines. We have added additional lines (137-141) to the manuscript to explain this process in detail. 

The statistical analysis section needs expanding. This reviewer cannot determine what analyses were applied for what purposes. For example, what “study variables” did the authors examine by Spearman Rank Correlation?

We have expanded on the statistics section. However, per reviewer 2 recommendations we have removed some portions of the statistical analysis performed.

The accuracy of short periods might be very dependent on specific dogs. Did the authors account for dog in their analysis? In other words agreement might be high for some dogs and low for others. Examining the % agreement by dog might reveal such a “between dog” effect.

We agree that % agreement to 24 hours mean HR is dog dependent. We have now added an additional summary figure (new fig 1) that highlights this information.

A further approach the authors should consider is to randomly select a 5-minute, 30-minute, 1 hour, 2-hour and 3-hour segment from each dog and see how often these randomly selected segments agree within the predefined limits. This would represent the clinical reality of a client obtaining a heart rate at home.

We did think about performing a random selection approach as recommended by the reviewer. One of the problems with such an approach is that a random selection of a short duration ECG recordings will result in inclusion of sampling periods that are not uniform among all dogs. For example, in one dog the sampling may occur during periods of rest while in others sampling may occur during extreme activity. We believe this would limit the usefulness of the direct clinical application of a shorter duration sampling. The authors believe that a better approach would be if prospective sampling was done under uniform conditions such as at rest or specific time of the day, as it would directly translate to clinical application. Unfortunately, given the retrospective nature of our study with incomplete Holter diary entries, we could not perform analysis limited to specific situations.

Minor comments:

Line 88-90. The authors state the limitations of Holters (cost, duration, compliance). However, the study examines Holter recordings and does not provide or discuss what alternatives to Holter monitoring exist or that could substitute Holter monitoring (and this reviewer knows of no such alternatives). Therefore, this statement seems rather contrived and without a solution.

Thank you for noting this. We understand how our statement is incomplete. We have added additional information clarifying our goal in the introduction and limitation sections. 

Lines 177-180. If the average rates are not different statistically, then one group cannot have a “slightly higher” rate than the other. Also, a P value should be provided for all statistical testing, whether significant or not. Please amend.

Requested changes made.

Line 177-180. The average HR for dogs with AF that were not being administered any rate control therapy seem considerably lower than previously documented in dogs with secondary AF. Do the authors have any explanation for this rather low rate? (most previous studies suggest rates exceeding 160-190 bpm). The rates for the group under rate control therapy appears to be consistent with previous observations.

We did notice such a discrepancy from previous studies. There are several reasons for the lower HR in the untreated group in this study. The data was edited to remove the initial portion of the Holter recording which is different compared to previous studies (only first 3 hours in the previous studies). To our knowledge, there is only one study where the 24-hour Holter mean HR was reported for the untreated group in the past (Gelzer AM, etal. Combination Therapy with Digoxin and Diltiazem Controls Ventricular Rate in Chronic Atrial Fibrillation in Dogs Better than Digoxin or Diltiazem Monotherapy: A Randomized Crossover Study in 18 Dogs). In the Gelzer study only dogs above a 24-mean HR of 140 bpm were included. Therefore, the true mean 24-hour HR in a group of untreated dogs is unknown. Additionally, in our institution Holter monitoring is routinely performed in most dogs to assess the need for rate control therapy prior to initiation of any rate control therapy. Similar to previous observations, we have noted in some of these dogs with extremely high in-hospital ECG HR that the at home 24-hour mean HR is low and does not warrant any rate control therapy. Therefore, portion of our untreated group is formed by such patients with low 24hour mean HR at home. We have added a few lines in the limitation’s sections to clarify this. 

Line 185-186. Could the authors provide scatter plots for each of the short duration recordings vs the Holter recordings plotted as average heart rates? Although the P value for their observation is “not significant”, with only 10 dogs in each group, the probability of this being a false negative is quite large.

We have removed the correlation analysis per reviewer 2 recommendations. From what we understand the reviewer is asking to provide scatter plot for pairs of each shorter duration ECG recording HR to 24 hours mean HR? It would be cumbersome to do. For example, for 5 min recordings there is 288 separate 5 min vs 24-hour plots needs to be done for each dog. Unless a specific 5 min sample is selected (the limitations of which we addressed in the previous comment) the utility of such a plot is limited. If the reviewer is asking for average of all 5 min HRs for each dog vs 24 hour mean HR scatter plot, then we believe that the average of all 5 min would be essentially same as the 24 hours mean HR.

Line 204-213. The authors suggest that a 5-minute ECG recording (in-clinic) would mostly agree to within 20% of the Holter-determined average HR. However, this is NOT what the authors showed. To demonstrate that specific idea, the authors would need to examine the 5 minute intervals within the 1st hour of Holter application (while the dog was still in the clinic and stressed) with the average at-home HR. How many of the 240 5-minute intervals obtained during the first hour came within 20% of the average at-home heart rate? If the authors suggest that clients could obtain a 5-minute ECG at home on their pets, the authors would have to suggest how such a feat would be possible, and also demonstrate that doing this would not alter the level of anxiety in the dog during at-home acquisition.

As the reviewers allude to there are several possible at home ECG monitoring devices that a client can use to obtain short term ECGs (AliveCor, cutaneous applied loop recorder (previously shown at our institution), patch recorders). While devices such as AliveCor requires owner-dog interaction the devices such as Cutaneous loop recorders can be activated to collect ECG with minimal interaction. With ever evolving technology there is likely further advances in ECG technology that would minimize the need for dog interaction thereby anxiety during data collection. We have added additional descriptions in introduction section and limitations section to clarify this.

Line 215-220. Several of the “health collars” (e.g. PetPace) can record ECGs in dogs, at least in sinus rhythm. Whether they can accurately count heart rates in dogs with atrial fibrillation remains to be determined. If they can, then they provide continuous monitoring, or can record variable periods for as long as desired (limited by battery life). Therefore, such devices (similar to the iWatch etc, used in humans) are available, but in limited supply. 

Agree that they are available and agree that they are currently untested. We have that research planned. 

Reviewer #2: General comments – this manuscript describes a retrospective analysis of subsets of ECG data recorded during a 24 hour Holter recording. While providing interesting preliminary data, I wonder how much less practical difficulty there might be in practice in obtaining shorter Holter recordings compared to a 24 hour recording? The dog still has to have the monitor fitted and wear it for several hours (preferably from the point the dog is relaxed at home, not from the initial time of Holter fitting) – if the benefits are “cost, recording duration and owner compliance” how much difference would shorter recordings truly make? I don’t have a good feel for how much less companies would charge to analyse shorter recordings, unfortunately. The main advantage I can see would be in those cases that the leads became detached after less than 24 hours – if it could be shown that the data was valid if at least 2-3 hours of recording was available once the dog was at home and relaxed that could prevent the need to start all over again. My specific comments, questions and observations are detailed below.

Dear reviewer, our goal with this study was not to replace a longer Holter recording with a shorter Holter recording. Rather we sought to assess the accuracy of a shorter ECG recording with available Holter data as a surrogate to shorter duration ECG devices for HR assessment in dogs with AF. While the data from this study could not be directly extrapolated to shorter duration device obtained HR information (due to factors such as human-dog interaction and such action on HR while collection). We have added additional information in introduction and limitations sections to further clarify this.

Introduction

Line 62 – should read “…remain a concern.”

Recommended changes made

Lines 63-65 – this sentence is identical to lines 59-61. I recommend deleting the sentence in lines 59-61, as it makes most sense to keep this information at the end of the paragraph.

Recommended changes made

Lines 67-79 – you mention that pulse assessment can be used to assess heart rate, but do not discuss later in the paragraph whether this would provide an accurate reflection of heart rate (which seems unlikely, due to pulse deficits, etc.).

Recommended addition made.

Lines 91-93 – how do you propose that the Holter is fitted without “the stress of hospital visits”?

See comments above

Results

Line 151 – just an idle observation, but I’m surprised only 20 dogs met the inclusion criteria over a 7 year period. Were there a lot of recordings that were <24 hours when edited?

Yes, the removal of the initial portion of the Holter recording resulted in less than 24 hours of Holter data in many AF dogs and therefore excluded from the study.

Lines 152-153 and 154-155 – personally, I would just report the range rather than the range and the IQR, as I don’t think the latter adds anything of real value.

Recommended changes made

Lines 177-180 – please don’t describe non-significant differences. This should simply read “There was no significant difference in heart rate between dogs receiving HR control therapy (121+/- 33bpm) and those not receiving HR control therapy (134 +/- 29 bpm).”

Recommended changes made.

Lines 180-182 – Table 1 and Figure 1 essentially show the same information twice. Please remove either the Table or the Figures.

Per recommendation the figure 1 is removed now.

Lines 182-184 – I’m not sure why this correlation is clinically interesting or important – please explain why you performed this test.

Lines 184-186 – given that rate control status had no significant impact on heart rate, why would you imagine that it would have an impact on accuracy? Again, this seems to be a statistical test that isn’t testing a hypothesis.

Excellent points; our hypothesis was that in dogs with higher mean 24-hour HR’s, the accuracy of shorter duration ECG will be better. We anticipated that dogs with higher 24-hour mean HR’s will have limited ability to reach lower HR ranges, and potentially less of a deviation from 24 hour mean HR in one direction resulting in narrower band width of shorter duration HR averages. But you are correct, since the HR was not different between two group the accuracy is unlikely to be different nor there likely be a relationship to the 24 hours mean HR. We have removed these analyses now. 

Table 1

The information displayed represents “Median (95% CI)” NOT “(Median +/- CI)” – please edit this accordingly.

Recommended correction made.

Discussion

Lines 206-207 – I don’t think that “agreement” changes – you are just being more lenient about what you are willing to accept as “accurate”.

We have made modifications to this statement.

Lines 209-215 – based on your stated mean heart rate of 126 bpm, a 20% inaccuracy gives a heart rate of 151 bpm, which is likely to be interpreted very differently than a mean of 126 by most clinicians. This is clearly potentially problematic – please expand the discussion to reflect this.

We have added additional statements as recommended.

Lines 215-220 – this is the first time you have mentioned alternative methods of obtaining an ECG recording in the home environment – which potentially changes ho you might be positioning shorter recordings (although I don’t relish the idea of asking a client to hold an AliveCor against their dog for an hour!). Please add this information to the Introduction (with a brief summary of the work that has already been done looking at accuracy of such recordings in the home environment for monitoring of AF).

Recommended description on previous work is now added to the introduction section.

Line 236 – please remove the grocer’s apostrophe (the “s” in ECGs denotes that this is plural rather than denoting possession).

Recommended change made

Lines 226-228 – is it not equally likely that these would consistently underestimate 24 hour averages (simply because they wouldn’t include the periods of exercise and excitement that also influence mean heart rate over 24 hours)? So you might be at risk of thinking that your rate control was better than it actually is?

Lines 237-248 – I am still not sure why you did this test in the first place (it feels like a fishing trip) so my preference would be to remove this from the discussion.

Lines 248-251 – this is possible, of course, but I think it’s very unlikely (see previous comment regarding lines 209-215).

These are great points. Taking your comments into consideration we have removed comments about prospective evaluations performed at resting ECGs.

Lines 255-257 – this would still rely on owners to record these times accurately – is there a way to mitigate this as a potential limitation to future studies?

We have removed this comment now per recommendation above. But in general, the owners would only be asked to record ECGs related to specific activities such as resting, sleeping or activity. With the time stamps from the owner recorded ECGs we can certainly match to a concurrently applied Holter recorder. 

Conclusions

Lines 260-262 – surely there is experimental evidence that would help determine guidelines for what is acceptable accuracy, rather than simply relying on the judgement of the clinician?

While there are some evidence on what an acceptable HR for an AF dog is, the authors are unaware of any studies that specifically looked at an acceptable accuracy for a ECG device for measuring HR in dogs with AF. In veterinary patients of AF we believe that the clinician judgement of an accuracy is likely vary and as such we would like to leave the conclusions statements in the current version. 

Reviewer #3: The purpose of this study was to determine if the heart rate measured for shorter time periods (5 minutes, one, two and three hours) would equate to the mean rate as determined by 24 hour electrocardiographic monitoring. The mean heart rate for each of these four time intervals was assessed for its ‘closeness’ (within 10%, 15% or 20%) to the 24 hour mean rate. The authors embarked on this investigation with the justification that performing a 24 hour Holter for shorter periods of time would be less problematic for owners and dogs. The conclusion of this paper is that none of the shorter time periods would equate to the mean 24-hour heart rate, but the longer the recording the closer the short surveillance heart rate was to the full recording.

This is a very focused study to determine if short recordings are adequate in the evaluation of rate during atrial fibrillation. The authors state that there are no other methods currently available to determine heart rates in a dog for longer than one hour. However, an Internet search will reveal that individuals are adapting noninvasive monitoring such as Fit Bits for dogs. Additionally, other devices have been developed. (See reference: Yasin Cotur, Michael Kasimatis, Matti Kaisti, Selin Olenik, Charis Georgiou, Firat Güder. Stretchable Composite Acoustic Transducer for Wearable Monitoring of Vital Signs. Adv. Funct. Mater. 2020, 30, 1910288. Doi.org/10.1002/adfm.201910288) These may serve as a way to get minimal information (and likely inadequate when monitoring arrhythmias) regarding heart rate. Such devices could provide data for specific questions such as heart rate. Comparison of these devices with established methods such as 24 hour Holter monitoring would be important to document. It is not clear how this study could be used in such a situation. Authors need to further justify this study

This comment is similar to previous reviewers comments we have clarified our goals and added additional statements to introduction and limitations sections.

From an implementation component there are no striking errors in this manuscript. It should be noted however that the average heart rates of the dogs examined are actually low for atrial fibrillation. Dogs that have high ventricular response rates are of course the more critical for monitoring and this paper does not address that kind of patient. Differences between low and high ventricular response rate heart rates over 24 hours is likely meaningful.

We have added additional explanations for why our 24hour mean HR was considerably lower in this study to the limitations section.

---

## [Decision Letter · Decision Letter 1]

4 Sep 2020

PONE-D-20-23023R1

Comparison of heart rate obtained from shorter duration Holter recordings to 24-hour mean heart rate in dogs with atrial fibrillation

PLOS ONE

Dear Dr. Sanders,

Thank you for submitting your manuscript to PLOS ONE. After careful consideration, we feel that it has merit but does not fully meet PLOS ONE’s publication criteria as it currently stands. Therefore, we invite you to submit a revised version of the manuscript that addresses the points raised during the review process.

Your manuscript was sent back to 2 of the previous Reviewer's who, together with myself, appreciate the modifications you have made to the the original manuscript. However, there are still a number of outstanding questions that should be resolved, particularly regarding statistical analysis.

We look forward to receiving your revised manuscript.

Kind regards,

Daniel M. Johnson, PhD

Academic Editor

PLOS ONE

Reviewers' comments:

Reviewer's Responses to Questions

**Comments to the Author**

1. If the authors have adequately addressed your comments raised in a previous round of review and you feel that this manuscript is now acceptable for publication, you may indicate that here to bypass the “Comments to the Author” section, enter your conflict of interest statement in the “Confidential to Editor” section, and submit your "Accept" recommendation.

Reviewer #1: (No Response)

Reviewer #2: (No Response)

2. Is the manuscript technically sound, and do the data support the conclusions?

Reviewer #1: (No Response)

Reviewer #2: Yes

3. Has the statistical analysis been performed appropriately and rigorously? 

Reviewer #1: (No Response)

Reviewer #2: Yes

4. Have the authors made all data underlying the findings in their manuscript fully available?

Reviewer #1: (No Response)

Reviewer #2: Yes

5. Is the manuscript presented in an intelligible fashion and written in standard English?

Reviewer #1: (No Response)

Reviewer #2: Yes

6. Review Comments to the Author

Reviewer #1: The authors have addressed most of my concerns. The only concern that I have is that individual dogs might have had "better agreement" than other dogs. A more complex analysis, where "dog" is inserted as a random variable into the model, might help demonstrate this.

To highlight this, one dog might have had agreement within 10% for 70% of the 5 minute segments, while another dog (for whatever reason) might have had agreement within 10% for only 10% of the 5 minute segments. A dog with an average AF rate of 230 bpm would have a higher probability of agreeing than one that still had considerable vagal influence on the AV nodal conduction.

This type of approach will likely require consultation with a statistician/epidemiologist. The authors might consider plotting the % agreement for each shorter period against average 24hr HR to see if there is any relationship.

Reviewer #2: General comments: thank you for the opportunity to read this revised version of the manuscript – I would like to thank the authors for addressing the majority of my previous comments and concerns. I have some remaining questions, which are detailed below.

Abstract

Line 40-41 – you say in lines 37-38 that “nearly 100% of heart rate averages obtained from 2 hour ECG recordings fell within 20% range” – but here you says that all of them did – please clarify this inconsistency.

Materials and Methods

Lines 158-159 – I’m still not sure why you are reporting both the range and the IQR. It seems superfluous to me – I would just report the range.

Lines 161-163 – it is not clear from this whether you mean “within 10% of the mean” (i.e. mean +/- 5%) or “mean+/- 10%” (which is what it says in the abstract) – please reword this to improve clarity.

Lines 161-163 – I assume that you mean that you are reporting the percentage of HR averages falling within 10%, 15% and 20% of the 24 hours mean HR FOR THAT DOG (rather than the average mean HR for all dogs)? Please clarify.

Results

Line 174 – again, I don’t know why you are reporting both range and IQR.

Lines 198-200 – again, the “10% range” is not the same as “mean +/- 10%”; it is mean +/- 5%. Please make sure that the wording is clear throughout the manuscript.

Lines 198-201 – an additional way to look at this would be to report the percentage of measurements that would have resulted in a different clinical decision being taken. For example, if (for the purposes of the study) we take 125bpm as a cut-off for deciding whether or not to change medications, you could report what percentage of the shorter recording were above this cut off if the 24 hour mean was <125 (and vice versa). This would be easier for clinicians to interpret, I think.

Table 1 – the poor accuracy for 3 hour recordings in the 10% range column is intriguing – the range includes 50% (i.e. no better than flipping a coin) – can you comment on the possible reasons for this apparent anomaly?

Discussion

Lines 232-234 – I appreciate what you are saying here (clearly, 20% of a larger number is bigger than 20% of a smaller number). However, I still think that even at lower heart rates 20% is too inaccurate – even at 120bpm, if you are talking about +/20% (which remains unclear – see previous comments) then this would go as high as 144bpm (which would almost certainly be interpreted differently). Even if you mean “within 20% of the mean” that would take you to 132bpm, which may well be interpreted differently from 120bpm. Personally, on the basis of your results I would want at least a 3 hour recording.

Lines 247-250 – I sincerely hope that clinicians do not accept wider limits of accuracy! See previous comments.

Lines 251-257 – these two sentences appear to contradict one another – if the other at home tools have the capacity to record continuous ECG for more than one hour, why is it not feasible to use them to make recording >1 hour long? Should it not say “…do not have the capacity to record continuous ECG for more than one hour duration”?

Conclusions

Lines 289-290 – that is a fair interpretation of your results – but I still maintain that the level of inaccuracy is potentially dangerous in terms of its impact on how the clinician might choose to manage the case (e.g. make dose adjustments to rate control medications (or not)). I think that it is very important that you acknowledge this overtly in the text – this is not simply a mathematical problem, but something that potentially impacts on how well patients are managed clinically.

Lines 290-292 – I think that it is really important that you provide more information that helps the reader understand better what 20% inaccuracy actually means for their patients; otherwise how can they judge what an acceptable level of error is?

7. PLOS authors have the option to publish the peer review history of their article (what does this mean?). If published, this will include your full peer review and any attached files.

Reviewer #1: No

Reviewer #2: No

---

## [Author Response · Author response to Decision Letter 1]

28 Sep 2020

Reviewer #1: The authors have addressed most of my concerns. The only concern that I have is that individual dogs might have had "better agreement" than other dogs. A more complex analysis, where "dog" is inserted as a random variable into the model, might help demonstrate this.

To highlight this, one dog might have had agreement within 10% for 70% of the 5 minute segments, while another dog (for whatever reason) might have had agreement within 10% for only 10% of the 5 minute segments. A dog with an average AF rate of 230 bpm would have a higher probability of agreeing than one that still had considerable vagal influence on the AV nodal conduction.

This type of approach will likely require consultation with a statistician/epidemiologist. The authors might consider plotting the % agreement for each shorter period against average 24hr HR to see if there is any relationship.

We have now added an additional analysis that accounts for the random effect of dogs and added necessary discussions. 

Reviewer #2: General comments: thank you for the opportunity to read this revised version of the manuscript – I would like to thank the authors for addressing the majority of my previous comments and concerns. I have some remaining questions, which are detailed below.

Abstract

Line 40-41 – you say in lines 37-38 that “nearly 100% of heart rate averages obtained from 2 hour ECG recordings fell within 20% range” – but here you says that all of them did – please clarify this inconsistency.

 “Nearly 100%” was used instead of “all” in the lines 37-38 since the CI for the median included 100%. We have changed the wording for consistency.

Materials and Methods

Lines 158-159 – I’m still not sure why you are reporting both the range and the IQR. It seems superfluous to me – I would just report the range.

Removed all applicable IQR throughout the manuscript

Lines 161-163 – it is not clear from this whether you mean “within 10% of the mean” (i.e. mean +/- 5%) or “mean+/- 10%” (which is what it says in the abstract) – please reword this to improve clarity.

Recommended changes made to improve clarity.

Lines 161-163 – I assume that you mean that you are reporting the percentage of HR averages falling within 10%, 15% and 20% of the 24 hours mean HR FOR THAT DOG (rather than the average mean HR for all dogs)? Please clarify.

Clarification provided

Results

Line 174 – again, I don’t know why you are reporting both range and IQR.

Recommended changes made

Lines 198-200 – again, the “10% range” is not the same as “mean +/- 10%”; it is mean +/- 5%. Please make sure that the wording is clear throughout the manuscript.

Recommended changes made throughout the manuscript

Lines 198-201 – an additional way to look at this would be to report the percentage of measurements that would have resulted in a different clinical decision being taken. For example, if (for the purposes of the study) we take 125bpm as a cut-off for deciding whether or not to change medications, you could report what percentage of the shorter recording were above this cut off if the 24 hour mean was <125 (and vice versa). This would be easier for clinicians to interpret, I think.

We do see practical utility in providing clinical decision cut-off and accuracy for such a cutoff above and below. The accuracy of a binary cutoff would be naturally better than accuracy for a specific percent range from 24-hour mean HR. For example, for a dog with a 24 hour mean HR of 200bpm, the 20% range would include heart rates from 140 to 240 bpm. If for instance this dog’s minimum and maximum heart rate ranges from 130 – 220 using 1-hour averages at home, 100% of the values will be above 125 bpm while a lot lesser percentage of values will be between 180-220. However, the practical utility may be limited given that a clinician may not know the predicted 24-hour mean HR at the time of any short-term ECG evaluation. We have highlighted this information as well in the discussion. As such given the limited practical utility we have not added this information.

Table 1 – the poor accuracy for 3-hour recordings in the 10% range column is intriguing – the range includes 50% (i.e. no better than flipping a coin) – can you comment on the possible reasons for this apparent anomaly?

There are at least two scenarios. One such scenario would be a dog with a high 24-hour mean HR and if such dog can reach really low HR values at rest. In contrast also can happen in dogs with lower mean HR but has the ability to reach extremely high max HR during activity. In either case, the analysis looks for HR values within an acceptable percentage from the 24 hour mean HR. The probability of falling out of that range may be lower than a coin toss in some dogs. If it a cutoff such as value above and below 125 bpm was used, then the probability (similar to a coin toss) would be expected to be above 50%. 

Discussion

Lines 232-234 – I appreciate what you are saying here (clearly, 20% of a larger number is bigger than 20% of a smaller number). However, I still think that even at lower heart rates 20% is too inaccurate – even at 120bpm, if you are talking about +/20% (which remains unclear – see previous comments) then this would go as high as 144bpm (which would almost certainly be interpreted differently). Even if you mean “within 20% of the mean” that would take you to 132bpm, which may well be interpreted differently from 120bpm. Personally, on the basis of your results I would want at least a 3 hour recording.

We agree that it is bit of a judgement call. We have now modified the sentence to better reflect what we are attempting to say.

Lines 247-250 – I sincerely hope that clinicians do not accept wider limits of accuracy! See previous comments.

That is our hope too. 

Lines 251-257 – these two sentences appear to contradict one another – if the other at home tools have the capacity to record continuous ECG for more than one hour, why is it not feasible to use them to make recording >1 hour long? Should it not say “…do not have the capacity to record continuous ECG for more than one hour duration”?

Yes it should say do not have the capacity. We apologize for the oversight. 

Conclusions

Lines 289-290 – that is a fair interpretation of your results – but I still maintain that the level of inaccuracy is potentially dangerous in terms of its impact on how the clinician might choose to manage the case (e.g. make dose adjustments to rate control medications (or not)). I think that it is very important that you acknowledge this overtly in the text – this is not simply a mathematical problem, but something that potentially impacts on how well patients are managed clinically.

We have added an additional statement to clarify this.

Lines 290-292 – I think that it is really important that you provide more information that helps the reader understand better what 20% inaccuracy actually means for their patients; otherwise how can they judge what an acceptable level of error is?

We have added an additional statement to provide clarification.

---

## [Decision Letter · Decision Letter 2]

7 Oct 2020

PONE-D-20-23023R2

Comparison of heart rate obtained from shorter duration Holter recordings to 24-hour mean heart rate in dogs with atrial fibrillation

PLOS ONE

Dear Dr. Sanders,

Thank you for submitting your manuscript to PLOS ONE. After careful consideration, we feel that it has merit but does not fully meet PLOS ONE’s publication criteria as it currently stands. Therefore, we invite you to submit a revised version of the manuscript that addresses the points raised during the review process.

Although the Reviewers and myself feel the changes you made much improved the manuscript, Reviewer 2 has a number of outstanding issues that need to be addressed before the manuscript can be accepted for publication in PLOS ONE.

We look forward to receiving your revised manuscript.

Kind regards,

Daniel M. Johnson, PhD

Academic Editor

PLOS ONE

Reviewers' comments:

Reviewer's Responses to Questions

**Comments to the Author**

1. If the authors have adequately addressed your comments raised in a previous round of review and you feel that this manuscript is now acceptable for publication, you may indicate that here to bypass the “Comments to the Author” section, enter your conflict of interest statement in the “Confidential to Editor” section, and submit your "Accept" recommendation.

Reviewer #1: All comments have been addressed

Reviewer #2: (No Response)

2. Is the manuscript technically sound, and do the data support the conclusions?

Reviewer #1: (No Response)

Reviewer #2: Yes

3. Has the statistical analysis been performed appropriately and rigorously? 

Reviewer #1: (No Response)

Reviewer #2: Yes

4. Have the authors made all data underlying the findings in their manuscript fully available?

Reviewer #1: (No Response)

Reviewer #2: Yes

5. Is the manuscript presented in an intelligible fashion and written in standard English?

Reviewer #1: (No Response)

Reviewer #2: Yes

6. Review Comments to the Author

Reviewer #1: (No Response)

Reviewer #2: General comments – thank you for responding to the majority of my previous comments and questions. I have a few remaining comments, which are detailed below.

Results

Lines 224-227 – surely the purpose of measuring the shorter duration heart rate averages is to try to predict the 24 hour mean heart rate; this analysis is therefore back-to-front.

Lines 227-229 – what is the P value for this effect?

Lines 229-232 – apologies if I have missed something, but I don’t understand where this information comes from – please can you explain how you derived this (i.e. where the information that “on…average 60% of the HR averages obtained from short duration ECG recordings fall within +/-10% of the 24 hour mean HR” comes from – I can’t figure it out from the preceding sentences).

Discussion

Lines 244-257 – I appreciate what you are trying to say – but this argument feels somewhat circular (essentially you can’t know what level of accuracy you need until you know what the 24 hour mean is, by which point you have done a 24h Holter, so the question becomes moot). Given this, can you suggest any situations in which a shorter recording WOULD be valuable? After all, surely the point of doing a 24h Holter in dogs with AF is that you don’t know what the 24h mean heart rate is and you want to find out? After all, even if you have previously Holtered a dog, surely the point of re-Holtering them is to see if anything has changed, so you don’t know what the mean HR is going to be before you do the Holter.

Conclusions

Lines 307-309 – I would still prefer this sentence to be edited or removed, as I don’t think that clinical functionality is determined by the clinician’s judgement (if something is wildly inaccurate that doesn’t change just because the clinician has decided that they are happy to accept a wildly inaccurate result).

Lines 309-314 – thank for adding this information – I think it provides very valuable context.

7. PLOS authors have the option to publish the peer review history of their article (what does this mean?). If published, this will include your full peer review and any attached files.

Reviewer #1: **Yes: **Mark Rishniw

Reviewer #2: No

---

## [Author Response · Author response to Decision Letter 2]

9 Oct 2020

Lines 224-227 – surely the purpose of measuring the shorter duration heart rate averages is to try to predict the 24 hour mean heart rate; this analysis is therefore back-to-front.

This analysis was requested by reviewer one to highlight the individual differences between dogs in accuracy of short duration ECG.

Lines 227-229 – what is the P value for this effect?

As such there is no p values associated with this effect. It was calculated as a % of individual variances (random effect) to total residual variance. We have modified the wording to reflect it. Prior to selecting the model, models with and without dog as a random effect was compared and addition of random effect was found to be statistically significant. 

Lines 229-232 – apologies if I have missed something, but I don’t understand where this information comes from – please can you explain how you derived this (i.e. where the information that “on…average 60% of the HR averages obtained from short duration ECG recordings fall within +/-10% of the 24 hour mean HR” comes from – I can’t figure it out from the preceding sentences).

The 60% in this sentence applies to the intercept of the linear mixed model.

Discussion

Lines 244-257 – I appreciate what you are trying to say – but this argument feels somewhat circular (essentially you can’t know what level of accuracy you need until you know what the 24 hour mean is, by which point you have done a 24h Holter, so the question becomes moot). Given this, can you suggest any situations in which a shorter recording WOULD be valuable? After all, surely the point of doing a 24h Holter in dogs with AF is that you don’t know what the 24h mean heart rate is and you want to find out? After all, even if you have previously Holtered a dog, surely the point of re-Holtering them is to see if anything has changed, so you don’t know what the mean HR is going to be before you do the Holter.

We appreciate your input on this. Prior to this publication, the accuracy of shorter duration ECG to predict 24 hour mean was unknown in AF dogs. If the shorter duration ECG predicted 24 hour mean HR better than the what was noted in this study, then certainly it could have replaced Holter monitoring for all dogs. But in this study the accuracy widely varied between dogs and was not influenced by the 24 hours mean HR, complicating further on selection of a desirable accuracy. We wanted to state this clearly to avoid any misleading information to the readers on the applicability of the results. As we pointed out in the discussion, one situation may be when the in-hospital ECG HR is above 155 bpm, then the 24-hour mean HR can be predicted to above 140 bpm. Then we would think 20% accuracy would be too high and a tighter accuracy is desired. 

Conclusions

Lines 307-309 – I would still prefer this sentence to be edited or removed, as I don’t think that clinical functionality is determined by the clinician’s judgement (if something is wildly inaccurate that doesn’t change just because the clinician has decided that they are happy to accept a wildly inaccurate result).

This sentence is removed 

Lines 309-314 – thank for adding this information – I think it provides very valuable context.

---

## [Decision Letter · Decision Letter 3]

19 Oct 2020

Comparison of heart rate obtained from shorter duration Holter recordings to 24-hour mean heart rate in dogs with atrial fibrillation

PONE-D-20-23023R3

Dear Dr. Sanders,

We’re pleased to inform you that your manuscript has been judged scientifically suitable for publication and will be formally accepted for publication once it meets all outstanding technical requirements.

Kind regards,

Daniel M. Johnson, PhD

Academic Editor

PLOS ONE

Additional Editor Comments (optional):

Reviewers' comments:

Reviewer's Responses to Questions

**Comments to the Author**

1. If the authors have adequately addressed your comments raised in a previous round of review and you feel that this manuscript is now acceptable for publication, you may indicate that here to bypass the “Comments to the Author” section, enter your conflict of interest statement in the “Confidential to Editor” section, and submit your "Accept" recommendation.

Reviewer #2: All comments have been addressed

2. Is the manuscript technically sound, and do the data support the conclusions?

Reviewer #2: Yes

3. Has the statistical analysis been performed appropriately and rigorously? 

Reviewer #2: Yes

4. Have the authors made all data underlying the findings in their manuscript fully available?

Reviewer #2: Yes

5. Is the manuscript presented in an intelligible fashion and written in standard English?

Reviewer #2: Yes

6. Review Comments to the Author

Reviewer #2: Thank you for your clear and comprehensive responses to my previous comments - I have no remaining questions.

7. PLOS authors have the option to publish the peer review history of their article (what does this mean?). If published, this will include your full peer review and any attached files.

Reviewer #2: No

---

## [Editor Report · Acceptance letter]

21 Oct 2020

PONE-D-20-23023R3 

Comparison of heart rate obtained from shorter duration Holter recordings to 24-hour mean heart rate in dogs with atrial fibrillation 

Dear Dr. Sanders:

I'm pleased to inform you that your manuscript has been deemed suitable for publication in PLOS ONE. Congratulations! Your manuscript is now with our production department. 

Kind regards, 

on behalf of

Dr. Daniel M. Johnson 

Academic Editor

PLOS ONE